# Subsampling scaling

A. Levina[1,2,*] & V. Priesemann[2,3,*]

In real-world applications, observations are often constrained to a small fraction of a system. Such spatial subsampling can be caused by the inaccessibility or the sheer size of the system, and cannot be overcome by longer sampling. Spatial subsampling can strongly bias inferences about a system's aggregated properties. To overcome the bias, we derive analytically a subsampling scaling framework that is applicable to different observables, including distributions of neuronal avalanches, of number of people infected during an epidemic outbreak, and of node degrees. We demonstrate how to infer the correct distributions of the underlying full system, how to apply it to distinguish critical from subcritical systems, and how to disentangle subsampling and finite size effects. Lastly, we apply subsampling scaling to neuronal avalanche models and to recordings from developing neural networks. We show that only mature, but not young networks follow power-law scaling, indicating self-organization to criticality during development.

[1] Institute of Science and Technology Austria, Am Campus 1, 3400 Klosterneuburg, Austria. [2] Bernstein Center for Computational Neuroscience, Am Fassberg 17, 37077 Göttingen, Germany. [3] Max Planck Institute for Dynamics and Self-Organization, Am Fassberg 17, 37077 Göttingen, Germany. * These authors contributed equally to this work. Correspondence and requests for materials should be addressed to A.L. (email: anna.levina@ist.ac.at) or to V.P. (email: viola@nld.ds.mpg.de).

Inferring global properties of a system from observations is a challenge, even if one can observe the whole system. The same task becomes even more challenging if one can only sample a small number of units at a time (spatial subsampling). For example, when recording spiking activity from a brain area with current technology, only a very small fraction of all neurons can be accessed with millisecond precision. To still infer global properties, it is necessary to extrapolate from this small sampled fraction to the full system.

Spatial subsampling affects inferences not only in neuroscience, but in many different systems: In disease outbreaks, typically a fraction of cases remains unreported, hindering a correct inference about the true disease impact[1,2]. Likewise, in gene regulatory networks, typically a fraction of connections remains unknown. Similarly, when evaluating social networks, the data sets are often so large that because of computational constraints only a subset is stored and analysed. Obviously, subsampling does not affect our inferences about properties of a single observed unit, such as the firing rate of a neuron. However, we are often confronted with strong biases when assessing aggregated properties, such as distributions of node degrees, or the number of events in a time window[3-6]. Concrete examples are distributions of the number of diseased people in an outbreak, the size of an avalanche in critical systems, the number of synchronously active neurons, or the number of connections of a node. Despite the clear difference between these observables, the mathematical structure of the subsampling problem is the same. Hence our novel inference approach applies to all of them.

Examples of subsampling biases, some of them dramatic, have already been demonstrated in numerical studies. For example, subsampling of avalanches in a critical model can make a simple monotonic distribution appear multi-modal[4]. In general, subsampling has been shown to affect avalanche distributions in various ways, which can make a critical system appear sub- or supercritical[5,7-11], and sampling from a locally connected network can make the network appear 'small-world'[6]. For the topology of networks, it has been derived that, contrary to common intuition, a subsample from a scale-free network is not itself scale-free[3]. Importantly, these biases are not due to limited statistics (which could be overcome by collecting more data, for example, acquiring longer recordings, or more independent subsamples of a system), but genuinely originates from observing a small fraction of the system, and then making inferences including unobserved parts. Although subsampling effects are known, in the literature there is so far no general analytical understanding of how to overcome them. For subsampling effects on degree distributions, Stumpf and colleagues provided a first analytical framework, stating the problem of subsampling bias[3].

In this paper, we show how to overcome subsampling effects. To this end we develop a mathematical theory that allows to understand and revert them in a general manner. We validate the analytical approach using various simulated models, and finally apply it to infer distributions of avalanches in developing neural networks that are heavily subsampled due to experimental constraints. Finally, we show that finite-size and subsampling effects clearly differ, and derived a combined subsampling-finite-size scaling relation. Together, our results introduce a novel approach to study under-observed systems.

## Results

**Mathematical subsampling.** To derive how spatial subsampling affects a probability distribution of observables, we define a minimal model of 'mathematical subsampling'. We first introduce the variables with the example of avalanches, which are defined as cascades of activity propagating on a network[12,13], and then present the mathematical definition. The main object of interest is a 'cluster', for example, an avalanche. The cluster size $s$ is the total number of events or spikes. In general, the cluster size is described by a discrete, non-negative random variable $X$. Let $X$ be distributed according to a probability distribution $P(X=s)=P(s)$. For subsampling, we assume for each cluster that each of its events is independently observed with probability $p$ (or missed with probability $1-p$). Then $X_{\text{sub}}$ is a random variable denoting the number of *observed* events of a cluster, and $X-X_{\text{sub}}$ the number of missed events. For neural avalanches, this subsampling is approximated by sampling a random fraction of all neurons. Then $X_{\text{sub}}$ represents the number of all events generated by the *observed* neurons within one avalanche on the full system. Note that this definition translates one cluster in the full system to exactly one cluster under subsampling (potentially of size zero; this definition does not require explicit binning, see Section 'Impact of binning' and Methods). We call the probability distribution of $X_{\text{sub}}$ 'subsampled distribution' $P_{\text{sub}}(s)$. An analogous treatment can be applied to, for example, graphs. There a 'cluster' represents the set of (directed) connections of a specific node, and thus $X$ is the degree of that node. Under subsampling, that is, considering a random subnetwork, only connections between *observed* nodes are taken into account, resulting in the subsampled degree $X_{\text{sub}}$.

As each event is observed independently, the probability of $X_{\text{sub}}=s$ is the sum over probabilities of observing clusters of $X=s+k$ events, where $k$ denotes the missed events and $s$ the sampled ones (binomial sampling):

$$P_{\text{sub}}(s)=P(X_{\text{sub}}=s)=\sum_{k=0}^{\infty} P(s+k)\binom{s+k}{s}p^s(1-p)^k. \quad (1)$$

This equation holds for any discrete $P(s)$ defined on $\mathbb{N}_0$, the set of non-negative integers. To infer $P(s)$ from $P_{\text{sub}}(s)$, we develop in the following a novel 'subsampling scaling' that allows to parcel out the changes in $P(s)$ originating from spatial subsampling. A correct scaling ansatz collapses the $P_{\text{sub}}(s)$ for any sampling probability $p$.

In the following, we focus on subsampling from two specific families of distributions that are of particular importance in the context of neuroscience, namely exponential distributions $P(s)=C_\lambda e^{-\lambda s}$ with $\lambda>0$, and power laws $P(s)=C_\gamma s^{-\gamma}$ with $\gamma>1$. These two families are known to show different behaviours under subsampling[3]:

1. For exponential distributions, $P(s)$ and $P_{\text{sub}}(s)$ belong to the same class of distributions, only their parameters change under subsampling. Notably, this result generalizes to positive and negative binomial distributions, which include Poisson distributions.
2. Power-laws or scale-free distributions, despite their name, are not invariant under subsampling. Namely, if $P(s)$ follows a power-law distribution, then $P_{\text{sub}}(s)$ is not a power law but only approaching it in the limit of large cluster size ($s\to\infty$).

In more detail, for exponential distributions, $P(s)=C_\lambda e^{-\lambda s}$, $s\in\mathbb{N}_0$, subsampling with probability $p$ results in an exponential distribution with decay parameter $\lambda_{\text{sub}}$ that can be expressed as a function of $\lambda$ and $p$ (for the full analytical derivation see Supplementary Note 1: Subsampling of negative binomial and exponential distributions):

$$\lambda_{\text{sub}}=\ln\left(\frac{e^\lambda+p-1}{p}\right) \Leftrightarrow \lambda=\ln\left((e^{\lambda_{\text{sub}}}-1)p+1\right). \quad (2)$$

Likewise, changes in the normalizing constant $C_\lambda=1-e^{-\lambda}$ of

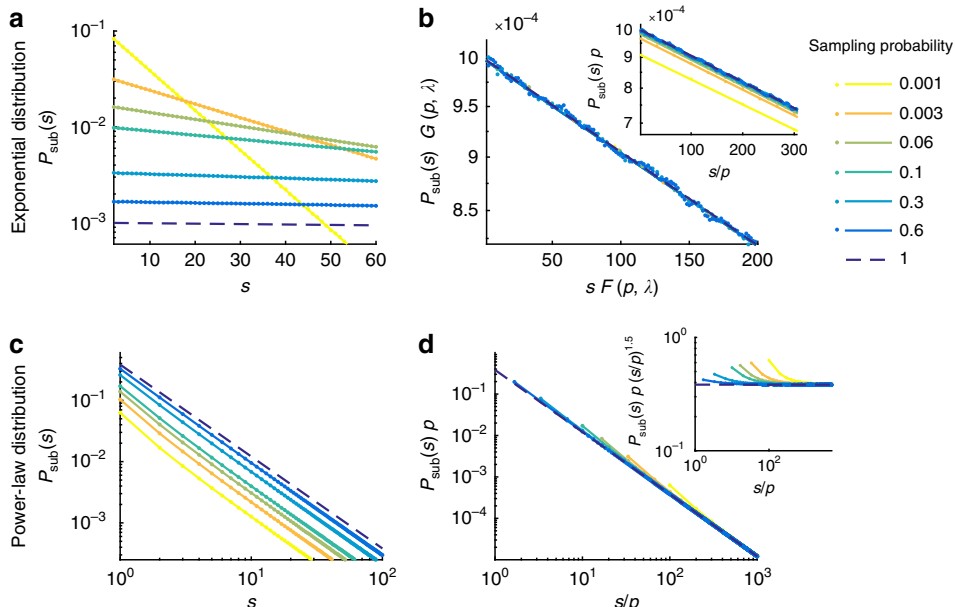

**Figure 1 | Mathematical subsampling of exponential and power-law distributions.** (**a**) Subsamplings of an exponential distribution with exponent $\lambda = 0.001$. (**b**) Collapse of subsampled exponential distributions by subsampling scaling derived in equation (4). Inset: same with p-scaling (equation (6)). (**c**) Subsampled power-law distributions with exponent $\gamma = 1.5$. (**d**) Collapse of the same distributions by p-scaling (equation (6)); inset: flattened version. Note the log-linear axes in **a**,**b**, and the double-logarithmic axes in **c**,**d**. Solid lines are analytical results (equation (1)), dots are numerical results from subsampling $10^7$ avalanches (realizations of the random variable $X$) of the corresponding original distribution. Colours indicate the sampling probability $p$.

$P(s)$ are given by:

$$C_\lambda / C_{\lambda_{\text{sub}}} = 1 - e^{-\lambda} + p e^{-\lambda} = \frac{e^{-\lambda_{\text{sub}}} + p - p e^{-\lambda_{\text{sub}}}}{p}. \quad (3)$$

These two relations allow to derive explicitly a subsampling scaling for exponentials, that is, the relation between $P(s)$ and $P_{\text{sub}}(s)$:

$$
\begin{aligned}
P(s) &= \frac{C_\lambda}{C_{\lambda_{\text{sub}}}} P_{\text{sub}}\left(\frac{\lambda}{\lambda_{\text{sub}}} s\right) \\
&= \frac{e^{-\lambda_{\text{sub}}} + p - p e^{-\lambda_{\text{sub}}}}{p} P_{\text{sub}}\left(\frac{\ln\left(e^{\lambda_{\text{sub}}} p - p + 1\right)}{\lambda_{\text{sub}}} s\right) \\
&= \left(1 - e^{-\lambda} + p e^{-\lambda}\right) P_{\text{sub}}\left(\frac{\lambda}{\ln\left(\frac{e^\lambda + p - 1}{p}\right)} s\right) \\
&= G(p, \lambda) P_{\text{sub}}(s F(p, \lambda)).
\end{aligned}
\quad (4)
$$

Thus given an exponential distribution $P(s)$ of the full system, all distributions under subsampling can be derived. Vice versa, given the observed subsampled distribution $P_{\text{sub}}(s)$, the full distribution can be analytically derived if the sampling probability $p$ is known. Therefore, for exponentials, the scaling ansatz above allows to collapse all distributions obtained under subsampling with any $p$ (Fig. 1a,b).

The presented formalism is analogous to the one proposed by Stumpf *et al.*[3]. They studied which distributions changed and which preserved their classes under subsampling. In the following we extend that study, and then develop a formalism that allows to extrapolate the original distribution from the subsampling, also in the case where an exact solution is not possible.

For power-law distributions of $X$, $X_{\text{sub}}$ is not power-law distributed, but only approaches a power law in the tail ($s \to \infty$). An approximate scaling relation, however, collapses the tails of distributions as follows (mathematical derivation in Supplementary Note 1: Subsampling of power-law distributions). For

$s \to \infty$, a power law $P(s) = C_\gamma s^{-\gamma}$ and the distributions obtained under subsampling can be collapsed by:

$$P(s) = p^a P_{\text{sub}}\left(p^b s\right), \text{ for any } a, b \in \mathbb{R} \text{ with } a - b\gamma = 1 - \gamma. \quad (5)$$

For any $a,b$ satisfying the relation above, this scaling collapses the tails of power-law distributions. The 'heads', however, deviate from the power law and hence cannot be collapsed (see deviations at small $s$, Fig. 1d). These deviations decrease with increasing $p$, and with $\gamma \to 1^+$ (ref. 3, Supplementary Note 1: Power-law exponent close to unity). We call these deviations '*hairs*' because they 'grow' on the 'heads' of the distribution as opposed to the tails of the distribution. In fact, the hairs allow to infer the system size from knowing the number of sampled units if the full systems exhibit a power-law distribution (Supplementary Note 1: Inferring the system size from the subsampled distribution).

In real-world systems and in simulations, distributions often deviate from pure exponentials or pure power laws[14,15]. We here treat the case that is typical for finite size critical systems, namely a power law that transits smoothly to an exponential around $s = s^{\text{cutoff}}$ (for example, Fig. 1a). Under subsampling, $s^{\text{cutoff}}_{\text{sub}}$ depends linearly on the sampling probability: $s^{\text{cutoff}}_{\text{sub}} = p \cdot s^{\text{cutoff}}$. Hence, the only solution to the power-law scaling relation (equation (5)) that collapses (to the best possible degree), both, the power-law part of distributions *and* the onsets of the cutoff is the one with $a = b = 1$:

$$P(s) \approx p P_{\text{sub}}(p \cdot s). \quad (6)$$

As this scaling is linear in $p$, we call it p-scaling. A priori, p-scaling is different from the scaling for exponentials (equation (4)). However, p-scaling is a limit case of the scaling for exponentials under the assumption that $\lambda \ll p$: Taylor expansion around $\lambda = 0$ results in the scaling relation $P(s) \approx p P_{\text{sub}}(p \cdot s)$, that is, the same as derived in equation (6). Indeed, for exponentials with $\lambda = 0.001$ p-scaling results in a nearly perfect collapse for all $p > 0.01$, however $p \leq 0.01$ violates the $\lambda \ll p$ requirement

and the collapse breaks down (Fig. 1b, inset). Thus p-scaling collapses power laws with exponential tail if $\lambda$ is small, and also much smaller than the sampling probability. This condition is typically met in critical, but not in subcritical systems (see Supplementary Note 2: Subcritical systems).

**Subsampling in critical models.** Experimental conditions typically differ from the idealized, mathematical formulation of subsampling derived above: Distributions do not follow perfect power laws or exponentials, and sampling is not necessarily binomial, but restricted to a fixed set of units. To mimic experimental conditions, we simulated avalanche generating models with fixed sampling in a critical state, because at criticality, subsampling effects are expected to be particularly strong: In critical systems, avalanches or clusters of activated units can span the entire system and thus under subsampling they cannot be fully assessed.

We simulated critical models with different exponents of $P(s)$ to assess the generality of our analytically derived results. The first model is the widely used branching model (BM)[8,16–19], and the second model is the Bak–Tang–Wiesenfeld model (BTW)[13], both studied in two variants. Both models display avalanches of activity after one random unit (neuron) has been activated externally (drive). In the BM, activity propagates stochastically, that is, an active neuron activates any of the other neurons with a probability $p_{act}$. Here $p_{act}$ is the control parameter, and the model is critical in the infinite size limit if one spike on average triggers one spike in its postsynaptic pool (see Methods). We simulated the BM on a fully connected network and on a sparsely connected network. The avalanche size distributions of both BM variants have an exponent $\approx 1.5$ (ref. 17), and for both variants, subsampling results are very similar (Supplementary Note 3: Subsampling of the EHE-model and sparse branching model). Hence in the main text we show results for the fully connected BM, while the results for the sparsely connected BM are displayed, together with results of a third model, the non-conservative model from Eurich, Herrmann and Ernst (EHE-model)[20], in Supplementary Note 3: Subsampling of the EHE-model and sparse branching model. As expected, distributions of all critical models collapse under p-scaling.

In the BTW, activity propagates deterministically via nearest neighbours connections. Propagation rules reflect a typical neural non-leaky 'integrate-and-fire' mechanism: Every neuron sums (integrates) its past input until reaching a threshold, then becomes active itself and is reset. The BTW was implemented classically with nearest neighbour connections on a 2D grid of size $M = L \times L$ either with open (BTW), or with circular (BTWC) boundary conditions. For the BTW/BTWC the exponent of $P(s)$ depends on the system size, and for the size used here ($M = 2^{14}$) it takes the known value of $\approx 1.1$ (ref. 21). Thus the slope is flatter than 1.29, which is expected for the infinite size BTW[21,22].

For subsampling, $N$ units were pre-chosen randomly. This subsampling scheme is well approximated by binomial subsampling with $p = N/M$ in the BM, because the BM runs on a network with full or annealed connections, and hence units are homogeneously connected. In the BTW/BTWC, subsampling violates the binomial sampling assumption, because of the models' deterministic, local dynamics.

For all models, the avalanche distributions under full sampling transit from an initial power law to an exponential at a cutoff $s^{cutoff} \approx M$ due to finite size effects (Fig. 2a). For small $s$, the hairs appear in the BM, originating from subsampling power laws (Fig. 2b, see Fig. 5a for a flattened version). These hairs are almost absent in the BTW/BTWC, because the power-law slope is close to unity (Supplementary Note 1: Power-law exponent close to

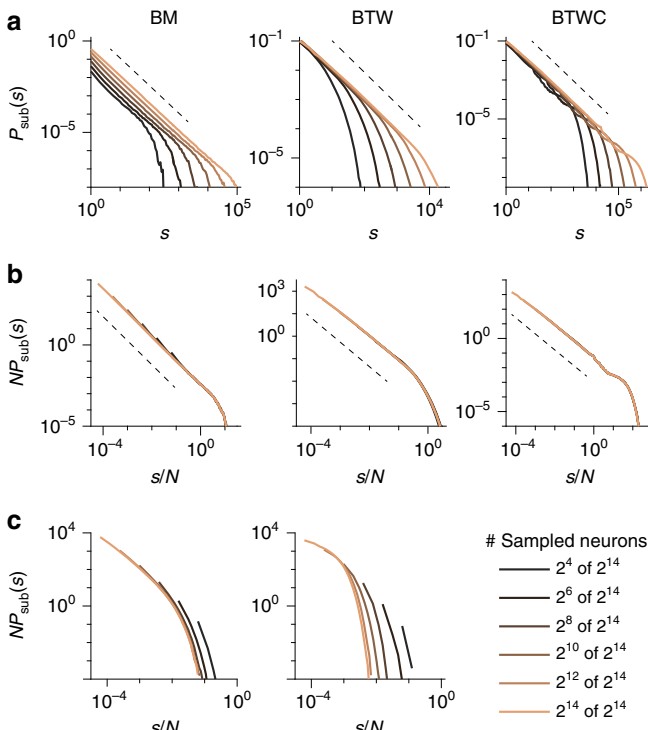

**Figure 2 | Subsampling scaling in critical and subcritical models.**
The three columns show results for the branching model (BM), the Bak–Tang–Wiesenfeld model (BTW), and the BTW with periodic boundary conditions (BTWC). (**a**) Avalanche size distribution $P_{sub}(s)$ for different degrees of subsampling, as denoted in the legend. (**b**) Same distributions as in **a**, but with p-scaling. (Note that scaling by $N$ leads to a collapse equivalent to scaling by $p = N/M$ at fixed system size $M$). (**c**) Scaled distributions from subcritical versions of the models. Here, results for the BTWC are extremely similar to those of the BTW and are thus omitted. Dashed lines indicate power-law slopes of $-1.5$ and $-1.1$ for the BM and BTW/BTWC, respectively, for visual guidance.

unity). The tails, even those of the BTWC, which have an unusual transition at the cutoff, collapse well. The BTWC is an exception in that it has unusual finite size effects, translating to the characteristic tails of $P(s)$. In fact, here the tails collapse better when applying fixed instead of binomial subsampling. (Fixed subsampling refers to pre-choosing a fixed set of units to sample from; this may violate mean-field assumptions.) This is because loosely speaking, binomial subsampling acts as a low pass filter on $P(s)$, smearing out the peaks, while fixed subsampling conserves the shape of the tails better here, owing to the compactness of the avalanches specifically in the 2D, locally connected BTWC. Overall, despite the models' violation of mean-field assumptions, the analytically motivated p-scaling ansatz allows to infer $P(s)$ from subsampling, including the detailed shapes of the tail.

**Distinguishing critical from subcritical systems.** Distinguishing between critical and subcritical systems under subsampling is particularly important when testing the popular hypothesis that the brain shows signatures of 'critical dynamics'. Criticality is a dynamical state that maximizes information processing capacity in models, and therefore is a favourable candidate for brain functioning[18,23–25]. Typically, testing for criticality in experiments is done by assessing whether the 'neural avalanche' distributions follow power laws[12]. Here, subsampling plays

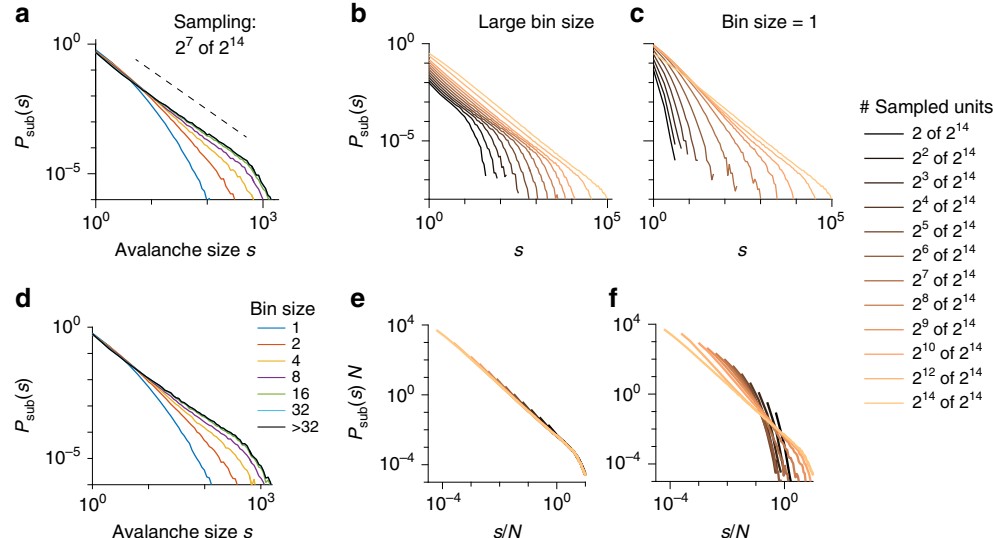

**Figure 3 | Impact of binning on avalanche size distributions $P_{sub}(s)$ and scaling.** (**a,d**) Sampling $N = 2^7$ units at different bin sizes from sparse (**a**) and fully connected network (**d**). For small bin sizes (<16 steps), $P_{sub}(s)$ deviates from a power law with slope 1.5 (dashed line). For larger bin sizes ($\geq$ 32 steps), $P_{sub}(s)$ is bin size invariant and shows the expected power law with cutoff. (**b,e**) Same as Fig. 2; for sufficiently large bin sizes $P_{sub}(s)$ collapsed under subsampling scaling. (**c,f**) When applying a small bin size, here one step, $P_{sub}(s)$ does not collapse. Parameters: Critical branching model (BM) with size $M = 2^{14}$ and sparse connectivity ($k = 4$), except for **d** that has all-to-all connectivity ($k = M$).

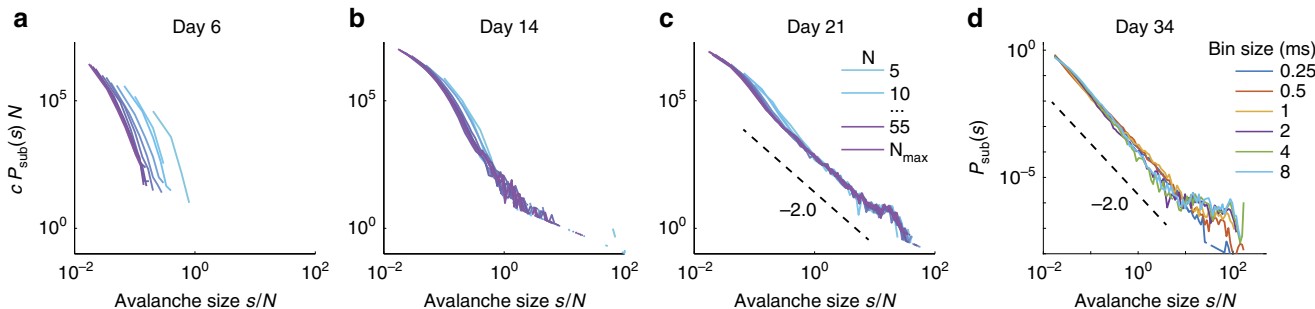

**Figure 4 | Avalanche size distributions $c \cdot P_{sub}(s)$ (in absolute counts) of spiking activity of developing neural networks *in vitro*.** (**a**) For young cultures, $P_{sub}(s)$ did not collapse under p-scaling, indicating that the full network does not show a power-law distribution for $P_{sub}(s)$. (**b,c**) More mature networks show a good collapse, allowing to extrapolate the distribution of the full network (see Supplementary Fig. 7) for all recording days of all experiments). In **a–c** the bin size is 1 ms, and $c$ is a total number of recorded avalanches in the full system, in **a**: $c = 53,803$, in **b**: $c = 307,908$, in **c**: $c = 251,156$. The estimated number of neurons in the cultures is $M \approx 50,000$. (**d**) $P_{sub}(s)$ from sampling spikes from all electrodes but evaluated with different bin sizes (see legend); the approximate invariance of $P_{sub}(s)$ against changes in the bin size indicates a separation of time scales in the experimental preparation.

a major role, because at criticality avalanches can propagate over the entire network of thousands or millions of neurons, while millisecond precise sampling is currently constrained to about 100 neurons. Numerical studies of subsampling reported contradictory results[4,5,8–10]. Therefore, we revisit subsampling with our analytically derived scaling, and compare scaling for critical and subcritical states.

In contrast to critical systems, *subcritical* ones lack large avalanches, and the cutoff of the avalanche size distribution is independent of the system size (if $M$ is sufficiently large). As a consequence, the distributions obtained under subsampling do not collapse under p-scaling. In fact, there exists no scaling that can collapse all subsampled distributions (for any $p$) simultaneously, as outlined below, and thereby p-scaling can be used to distinguish critical from non-critical systems.

The violation of p-scaling in subcritical systems arises from the incompatible requirement for scaling at the same time the power-law part, the exponential tail and the cutoff onset $s_{sub}^{cutoff}$. On the one hand, the exponential tail becomes increasingly steeper with distance from criticality (larger $\lambda$), so that the relation $\lambda \ll p$ required for p-scaling (equation (6)) does not hold anymore for small $p$ (more details in Supplementary Note 2: Subcritical systems). Thus, a collapse of the tails would require the scaling ansatz for exponentials (equation (4)). On the other hand, slightly subcritical models still exhibit power-law behaviour up to a cutoff $s^{cutoff} := c$ that is typically much smaller than the system size ($c \ll M$). To properly scale this part of the distribution, p-scaling is required. Likewise, the onset of the cutoff scales under subsampling with $p$: $s_{sub}^{cutoff} = c \cdot p$, requiring a scaling of the $s$-axis in the same manner as in the p-scaling. Thus, because the exponential decay requires different scaling than the power law and $s_{sub}^{cutoff}$, no scaling ansatz can collapse the entire distributions from 'head to tail'.

**Impact of binning.** The main focus of this paper is to show how distributions of avalanches, node degrees or other 'clusters'

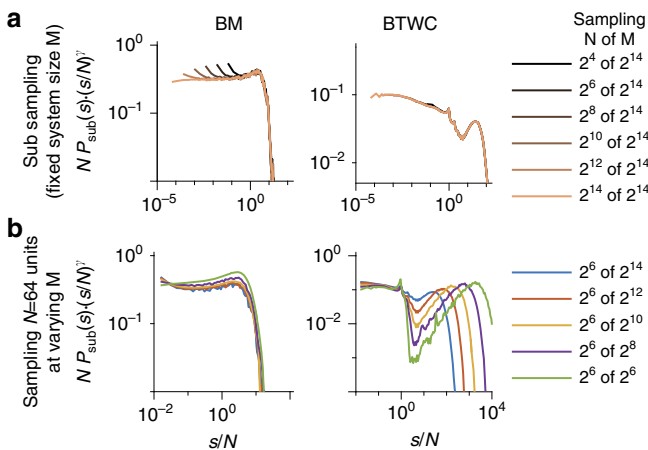

**Figure 5 | Changes in $P_{sub}(s)$ mediated by system size ($M$) and sampling size ($N$).** ($a$,$b$) Scaled and flattened avalanche size distribution ($P_{sub}(s)$) for the branching model (BM, left) and the Bak–Tang–Wiesenfeld model with circular boundary conditions (BTWC, right); flattening is achieved by multiplying $P_{sub}(s)$ with a power law with appropriate slope $\gamma$. We used $\gamma = 1.5$ and $\gamma = 1$ for the BM and BTWC, respectively. ($a$) $P_{sub}(s)$ for different samplings ($N = 2^4 \ldots 2^{14}$) from models with fixed size $M = 2^{14}$. Note the 'hairs' in the BM induced by subsampling. ($b$) $P_{sub}(s)$ from sampling a fixed number of $N = 2^6$ neurons from models of different sizes ($M = 2^6 \ldots 2^{14}$). Note the difference in distributions despite the same number of sampled neurons, demonstrating that finite size effects and subsampling effects are not the same.

change under spatial subsampling, and how to infer the distribution of the fully sampled system from the subsampled one. To this end, it is essential that the clusters are extracted unambiguously, that is, one cluster in the full system translates to exactly one cluster (potentially of size zero) under subsampling. This condition is easily realized for the degree of a node: One simply takes into account only those connections that are realized with other *observed* nodes. For avalanches, this condition can also be fulfilled easily if the system shows a separation of time scales (STS), that is, the pauses between subsequent avalanches are much longer than the avalanches themselves. With an STS, temporal binning[12] can be used to unambiguously extract avalanches under subsampling. However, the chosen bin size must neither be too small nor too large: If too small, a single avalanche on the full system can be 'cut' into multiple ones when entering, leaving and re-entering the recording set. This leads to steeper $P_{sub}(s)$ with smaller bin size (Fig. 3a). In contrast, if the bin size is too large, subsequent avalanches can be 'merged' together. For a range of intermediate bin sizes, however, $P_{sub}(s)$ is invariant to changes in the bin size. In Fig. 3a, the invariance holds for all bin sizes $32 < \text{bin size} < \infty$. The result does not depend on the topology of the network (compare Fig. 3a for a network with sparse topology and Fig. 3d for fully connected network). If a system, however, lacks an STS, then $P_{sub}(s)$ is expected to change for any bin size. This may underlie the frequently observed changes in $P_{sub}(s)$ in neural recordings[4,7,12,26–28], as discussed in ref. 8.

To demonstrate the impact of the bin size on p-scaling, we here used the BM, which has a full STS, that is, the time between subsequent avalanches is mathematically infinite. When sampling $N = 2^7$ out of the $M = 2^{14}$ units, then $P_{sub}(s)$ deviates from a power law for small bin sizes and only approaches a power law with the expected slope of 1.5 for bin sizes larger than eight steps (Fig. 3a). The same holds for subsampling of any $N$: With sufficiently large bin sizes, $P_{sub}(s)$ shows the expected

approximate power law (Fig. 3b). In contrast, for small bin sizes avalanches can be cut, and hence $P_{sub}(s)$ deviates from a power law (Fig. 3c). This effect was also observed in refs 8,9, where the authors used small bin sizes and hence could not recover power laws in the critical BM under subsampling, despite an STS. Thus in summary, p-scaling only collapses those $P_{sub}(s)$, where avalanches were extracted unambiguously, that is, a sufficiently large bin size was used (compare Fig. 3e,f).

The range of bin sizes for which $P_{sub}(s)$ is invariant depends on the specific system. For the experiments we analysed in the following section, we found such an invariance for bin sizes from 0.25 to 8 ms if $P_{sub}(s)$ follows a power law, indicating indeed the presence of an STS (Fig. 4d). Thus our choice of 1 ms bin size suggests an unambiguous extraction of avalanches and in this range p-scaling works as predicted theoretically.

**Subsampled neural recordings**. We applied p-scaling to neural recordings of developing networks *in vitro* to investigate whether their avalanches indicated a critical state. To this end, we evaluated recordings from $N = 58$ multi-units (see Methods[29]). This is only a small fraction of the entire neural network, which comprised $M \approx 50,000$ neurons; thus the avalanche size distribution obtained from the whole analysed data is already a subsampled distribution $P_{sub}(s)$. To apply p-scaling, we generated a family of distributions by further subsampling, that is, evaluating a subset $N' < N$ of the recorded units. In critical systems, p-scaling is expected to collapse this family of distributions if avalanches are defined unambiguously, as outlined above.

Interestingly, for early stages of neural development, p-scaling does not collapse $P_{sub}(s)$, but for the more mature networks we found a clear collapse (Fig. 4). Thus developing neural networks start off with collective dynamics that is not in a critical state, but with maturation approach criticality[30,31]. Some of the mature networks show small bumps in $P_{sub}(s)$ at very large avalanche sizes ($s \approx 5,000 \Leftrightarrow s/N \approx 60$). These very large avalanches comprise only a tiny fraction of all avalanches (about 2 in 10,000). At first glance, the bumps are reminiscent of supercritical systems. However, supercritical neural models typically show bumps at system or sampling size ($s = N$), not at those very large sizes. We demonstrate the data from all experiments and discuss these deviations in more detail in Supplementary Note 4: Detailed discussion of the experimental results, and suggest that the bumps are more likely to originate from neurophysiological finite size effects.

For the full, mature network, our results predict that $P(s)$ would extend not only over three orders of magnitude as here, but over six, because $p \approx 10^{-3}$. Our analysis of neural recordings illustrates how further spatial subsampling allows to infer properties of the full system, even if only a tiny fraction of its collective dynamics has been observed, simply by sampling even less ($N' < N$) of the full system.

**Subsampling versus finite size scaling**. In the real world we are often confronted with data affected by both subsampling and finite system size effects, that is, observations originated from a small part of a large, but not infinite system. Thus we need to deal with a combination of both: subsampling effects as a result of incomplete data acquisition and finite-size effects inherited from the full system. To disentangle influences from system size and system dynamics, finite size scaling (FSS) has been introduced[32,33]. It allows to infer the behaviour of an infinite system from a set of finite systems. At a first glance, finite size and subsampling effects may appear to be very similar. However, if they were, then distributions obtained from sampling $N$ units

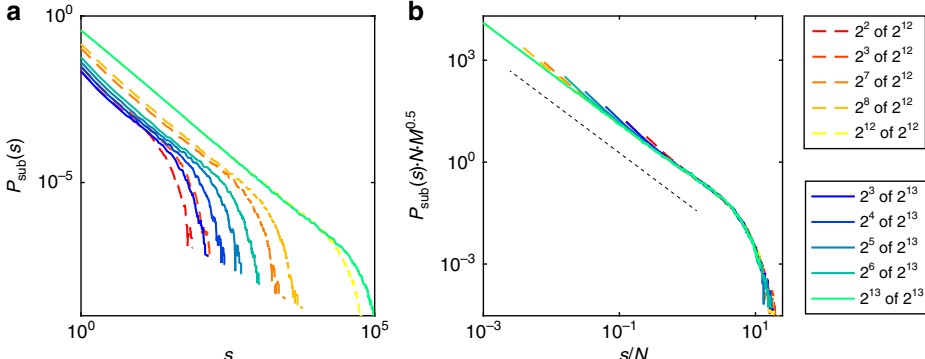

**Figure 6 | Subsampling scaling combined with finite-size scaling.** (**a**) Subsampling distributions for different numbers of sampled units $N$ from the BM with system sizes $M_1 = 2^{12}$ and $M_2 = 2^{13}$. (**b**) Distributions collapsed as predicted by applying subsampling-finite-size scaling (equation (7)), with $\gamma = 1.5$ and $v = 1$. The dashed black line indicates a slope of $-1.5$ for visual guidance.

from any system with $N \le M$ would be identical, that is, independent of $M$. This is not the case, as, for example, the distributions for fixed $N = 2^6$ clearly depend on $M$ (Fig. 5b). In fact, in both models the tails clearly inherit signatures of the full system size. Moreover, in the BM, subsampling a smaller fraction $p = N/M$ of a system increases the 'hairs', an effect specific to subsampling, not to finite size (see the increasing convexity of the flat section with decreasing $p$ in the BM, Fig. 5b).

Importantly, as shown above, for critical systems one can always scale out the impact of subsampling, and thereby infer the distribution of the full system, including its size-specific cutoff shape (Fig. 5a). Hence, it is possible to combine FSS and subsampling scaling (detailed derivations are in Supplementary Note 5: Combining subsampling scaling and finite-size scaling): Consider a critical system, where FSS is given by: $M^\beta P(sM^v; M) = g(s)$, here $g(s)$ is a universal scaling function. Then FSS can be combined with subsampling scaling to obtain a universal subsampling-finite-size scaling:

$$NM^{\beta-1}P_{\text{sub}}\left(sNM^{v-1}; M, N\right) = g(s). \qquad (7)$$

Using equation (7) allows to infer the distribution for arbitrary subsampling ($N$) of any system size ($M$), Fig. 6.

## Discussion

The present study analytically treats subsampling scaling for power laws (with cutoff), exponential distributions, and negative and positive binomial distributions. For all other distributions, utmost care has to be taken when aiming at inferences about the full system from its subsampling. One potential approach is to identify a scaling ansatz numerically, that is, minimizing the distance between the different $P_{\text{sub}}(s)$ numerically, in analogy to the approach for avalanche shape collapse[7,26,34–36]. We found that for our network simulations such a numerical approach identified the same scaling parameters as our analytic derivations (Supplementary Note 6: Numerical estimation of optimal scaling). However, given the typical noisiness of experimental observations, a purely numerical approach should be taken with a grain of salt, as long as it is not backed up by a circular form analytical solution.

Our analytical derivations assumed annealed sampling, which in simulations was well approximated by pre-choosing a random subset of neurons or nodes for sampling. Any sampling from randomly connected networks is expected to lead to the same approximation. However, in networks with, for example, local connectivity, numerical results depend strongly on the choice of sampled units[4]. For example, for windowed

subsampling (that is, sampling a local set of units) a number of studies reported strong deviations from the expected power laws in critical systems or scale-free networks[4–6]. In contrast, random subsampling, as assumed here for our analytical derivations, only leads to minor deviations from power laws (hairs). Thus to diminish corruption of results by subsampling, future experimental studies on criticality should aim at implementing random instead of the traditional windowed sampling, for example, by designing novel electrode arrays with pseudo-random placement of electrodes on the entire area of the network. In this case, we predict deviations from power laws to be minor, that is, limited to the 'hairs' and the cutoff.

We present here first steps towards a full understanding of subsampling. With our analytical, mean-field-like approach to subsampling we treat two classes of distributions and explore corresponding simulations. In future, extending the presented approach to a window-like sampling, more general forms of correlated sampling, and to further classes of distributions will certainly be of additional importance to achieve unbiased inferences from experiments and real-world observations.

## Methods

**Analytical derivations.** The analytical derivations are detailed in the Supplementary Information.

**Bak–Tang–Wiesenfeld model.** The BTW model[13] was realized on a 2D grid of $L \times L = M$ units, each unit connected to its four nearest neighbours. Units at the boundaries or edges of the grid have either 3 or 2 neighbours, respectively (open boundary condition). Alternatively, the boundaries are closed circularly, resulting in a torus (circular or periodic boundary condition, BTWC). Regarding the activity, a unit at the position $(x, y)$ carries a potential $z(x, y, t)$ at time $t$, $(z, t \in \mathbb{N}_0)$. If $z$ crosses the threshold of 4 at time $t$, its potential is redistributed or 'topples' to its nearest neighbours:

$$\text{if } z(x, y, t) \ge 4:$$
$$z(x, y, t+1) = z(x, y, t) - 4$$
$$z(x \pm 1, y \pm 1, t+1) = z(x \pm 1, y \pm 1, t) + 1$$

$z(x \pm 1, y \pm 1)$ refers to the 4 nearest neighbours of $z(x, y)$. The BTW/BTWC is in an absorbing (quiescent) state if $z(x, y) < 4$, for all $(x, y)$. From this state, an 'avalanche' is initiated by setting a random unit $z(x, y)$ above threshold: $z(x, y, t+1) = z(x, y, t) + 4$. The activated unit topples as described above and thereby can make neighbouring units cross threshold. These in turn topple, and this toppling cascade propagates as an avalanche over the grid until the model reaches an absorbing state. The size $s$ of an avalanche is the total number of topplings. Note that the BTW/BTWC are initialized arbitrarily, but then run for sufficient time to reach a stationary state. Especially in models with large $M$ this can take millions of time steps.

The BTW and the BTWC differ in the way how dissipation removes potential from the system. Whereas in the BTW potential dissipates via the open boundaries, in the BTWC an active unit is reset without activating its neighbours with a tiny

probability, $p_{dis} = 10^{-5}$. For BTW an additional dissipation in a form of small $p_{dis}$ can be added to make the model subcritical.

**Branching model.** The BM corresponds to realizing a classical branching process on a network of units[8,17,18]. In the BM, an avalanche is initiated by activating one unit. This unit activates each of the $k$ units it is connected to with probability $p_{act}$ at the next time step. These activated units, in turn, can activate units following the same principle. This cascade of activations forms an avalanche which ends when by chance no unit is activated by the previously active set of units. The control parameter of the BM is $\sigma = p_{act} \cdot k$. For $\sigma = 1$, the model is critical in the infinite size limit. We implemented the model with full connectivity ($k = M$) and with sparse, annealed connectivity ($k = 4$). The BM can be mathematically rigorously associated with activity propagation in an integrate and fire network[37,38].

For implementation of the BM with full connectivity ($k = M = 2^{14}$), note that the default pseudo-random number generator (PRNG) of Matlab(R) (R2015b) can generate avalanche distributions that show strong noise-like deviations from the expected power-law distribution. These deviations cannot be overcome by increasing the number of avalanches, but by specifying a different PRNG. We used the 'Multiplicative Lagged Fibonacci' PRNG for the results here, because it is fairly fast.

**Subcritical models.** To make the models subcritical, in the BM $\sigma$ was set to $\sigma = 0.9$, and in the BTW/BTWC the dissipation probability $p_{dis}$ was set to $p_{dis} = 0.1$, which effectively corresponds to $\sigma = 0.9$, because 90% of the events are transmitted, while 10% are dissipated.

**Avalanche extraction in the models.** The size $s$ of an avalanche is defined as the total number of spikes from the seed spike until no more units are active. Under subsampling, this translates to the total number of spikes that occur on the pre-chosen *set of sampled units* (fixed subsampling). In principle, the avalanches could also have been extracted using the common binning approach[12], as all the models were simulated with a separation of time scales (STS), that is, the time between subsequent avalanches is by definition much longer than the longest-lasting avalanche. Hence applying any bin size that is longer than the longest avalanche, but shorter than the pauses between avalanches would yield the same results for any subsampling.

**Data acquisition and analysis.** The spike recordings were obtained by Wagenaar et al.[29] from an *in vitro* culture of $M \approx 50,000$ cortical neurons. Details on the preparation, maintenance and recording setting can be found in the original publication. In brief, cultures were prepared from embryonic E18 rat cortical tissue. Recording duration of each data set was at least 30 min. The recording system comprised an $8 \times 8$ array of 59 titanium nitride electrodes with 30 μm diameter and 200 μm inter-electrode spacing, manufactured by Multichannel Systems (Reutlingen, Germany). As described in the original publication, spikes were detected online using a threshold-based detector as upward or downward excursions beyond 4.5 times the estimated RMS noise[39]. Spike waveforms were stored, and used to remove duplicate detections of multiphasic spikes. Spike sorting was not employed, and thus spike data represent multi-unit activity.

For the spiking data, avalanches were extracted using the classical binning approach as detailed in refs 8,12. In brief, temporal binning is applied to the combined spiking activity of all channels. Empty bins by definition separate one avalanche from the next one. The avalanche size $s$ is defined as the total number of spikes in an avalanche. The bin size applied here was 1 ms, because this reflects the typical minimal time delay between a spike of a driving neuron and that evoked in a monosynaptically connected receiving neuron, and because 1 ms is in the middle of the range of bin sizes that did not change the avalanche distribution $P_{sub}(s)$ (Fig. 4d).

Application of p-scaling by definition requires that one avalanche in the full system translates to one avalanche (potentially of size zero) under subsampling, that is, an avalanche must not be 'cut' into more than one, for example, when leaving and re-entering the recording set. This can be achieved in experiments that have a separation of time scales by applying a sufficiently large bin size, because this allows for an unambiguous avalanche extraction[8]. Indeed, the *in vitro* recordings we analyse here appear to show a separation of time scales: We found that varying the applied bin size around 1 ms hardly changed $P_{sub}(s)$ (Fig. 4d). In contrast, using too small bin sizes would have led to 'cutting' avalanches, which impedes the observation of power laws, and consequently prevents the collapse (illustrated for the BM, Fig. 3).

**Data availability.** We evaluated ten recordings for each day, because then the naïve probability of finding the expected behaviour consistently in all of them by chance is at most $p = (1/2)^{10} < 0.001$. The experimental data were made available online by the Potter group[29]. In detail, we downloaded from the dense condition the *in vitro* preparations 2-1, 2-3, 2-4, 2-5, 2-6, 6-1, 6-2, 6-3, 8-1, 8-2, 8-3, and for each preparation one recording per week (typically days 7, 14, 21, 28, 34/35, but for some experiments one or two days earlier), except for experiment 6-2 where we

only got the first three weeks, and 6-3 where we got the last two weeks. We analysed and included into the manuscript all recordings that we downloaded.

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

## Acknowledgements

The authors thank Georg Martius and Michael Wibral for inspiring discussions and support. We thank Jens Wilting, Johannes Zierenberg and João Pinheiro Neto for careful proof reading. A.L. received funding from the People Program (Marie Curie Actions) of the European Union's Seventh Framework Program (FP7/2007–2013) under REA grant agreement no. [291734]. V.P. received financial support from the Max Planck Society. A.L. and V.P. received financial support from the German Ministry for Education and Research (BMBF) via the Bernstein Center for Computational Neuroscience (BCCN) Göttingen under Grant No. 01GQ1005B.

## Author contributions

A.L. and V.P. contributed equally to the work. All results were developed jointly.

## Additional information

**Publisher's note**: 

