## [Peer review file · Nature Communications]

Reviewers' comments:

Reviewer #1 (Remarks to the Author):

In this manuscript, Levina and Priesemann aim to address the problem of subsampling, which has plagued the field of neuronal avalanches for many years. Priesemann has previously published on this, but the present work with Levina gives a firm analytical foundation to this problem and is therefore an important new contribution. Their results are convincing and demonstrate to my satisfaction that they have made great progress in this area, coming up with the correct equations to describe how power law and exponential distributions will be affected by subsampling. Based on this method, they go on to show that data collapse of the avalanche size distribution will occur only in models that are actually tuned to be critical, and that such collapse will not occur in models that are not critical. They also clearly demonstrate how the problem of subsampling is different from the problem of finite sized data. Finally, they apply their method to a publicly-available data set of cultured networks (from Potter's group) and show that mature cultures show collapse, while young and still developing cultures do not.

I think that this is a very important new contribution, and clearly merits publication in Nature Communications for two main reasons. First, this problem of subsampling has been known for about a decade, but no clear solution has been presented until now. Here, this controversy is nearly solved, at least for some of the main cases (power law and exponential distributions). Second, this area of criticality in biological systems is receiving increased attention and is not confined merely to neuronal avalanches. Criticality has been suggested in bacterial colonies, flocks of birds, and even human interactions. This work is thus very broadly applicable and is likely to be well-cited.

That said, I have a few comments, questions and suggestions:

The problem of subsampling that you address is confined to limited spatial sampling (e.g., not having enough electrodes to record all the neurons in the population). This is an enormous problem and tackling it is a commendable task. But could you at least say a few words about how this problem is related to two other issues? I am thinking of recording length and temporal binning. Your ability to properly correct for subsampling assumes that the data were collected over a long period of time (at least one hour), and this may be limited in many preparations (those that rely on Ca^{++} imaging, for example in neuroscience). Must recordings be of some length (assuming some firing rate) for all of this to work? Somewhat related to this, the temporal binning of the data you used was at 1 ms. But many labs bin at 5 ms, or even longer bins. Can you comment on how these will affect your conclusions on subsampling? In some sense, both of these issues are related to subsampling because they are also presenting limited views of what is actually going on in the data.

When doing data collapse, there are generally two ways that one could proceed. First, you can get the analytically expected parameters (as you did) and then show that they produce a reasonable collapse when inspected by eye. Second, you can conduct a collapse by minimizing some type of error after searching through different parameter values. In this manuscript, you did the former but not the latter. I see that your plots look "pretty good" but I would also like to see some type of independent corroboration of their quality. Can you show that deviating from the parameter values you have chosen will actually increase the distance between the curves? There are now some toolboxes out there that claim to do automated collapse by minimization of error.

Minor:

In supplementary figure 1, what is ASMC? Don't you mean BTWC? Or BTW with periodic boundary Conditions?

In the supplementary figures, it looks like there is a "supercritical bump" in some of the files, even though they show what appears to be good collapse. Can you explain this apparent discrepancy?

Reviewer #2 (Remarks to the Author):

I have read with interest the manuscript by A. Levina and V. Priesemann. It brings an interesting analytical understanding to the problem of subsampling, which has deservedly received considerable attention in the literature of neuronal avalanches. Specifically, the paper assumes that events X originally described by a probability distribution $P(X)$ are sampled with a probability p , rendering events X_{sub} whose effective probability distribution is $P_{\text{sub}}(X_{\text{sub}})$. Scaling relations relating P_{sub} to P and p are derived, some of which had been previously published by Stumpf et al PNAS 2005 (Ref. [3] of the manuscript). I find that the paper is relevant and mostly well written. There are, however, essential points that I would like to see discussed before the manuscript could be considered suitable for publication in Nature Communications.

To start with, I suggest the authors provide simple examples of the mathematical quantities upon which they are going to develop all their calculations (N_0 , N , X , X_{sub} etc). One can understand them in retrospect, once the calculations are advanced, but the manuscript could read much better and gain much more traction if the description of the problem was clearer from the start. Lines 59 and 64 have unclear expressions, such as “ N_0 that represents the number of events in a cluster”, or “each event or component of every cluster is independently observed”. Since a “cluster” has not been properly defined, the above sentences, as they stand, hinder more than they help. Perhaps one simple example preceding them would suffice.

Most importantly, the authors' mathematical description of subsampling leave aside, as far as I can understand, a very important aspect of subsampling, namely, the time-dependent nature of the underlying processes. Consider, for instance, neuronal avalanches, whose sizes are defined as the number of events (e.g. local field potential large-enough deviations) that occur in between silent epochs. To properly define what a silent epoch is, data is usually binned using some characteristic time of the system. In Ref. [20], for instance, this characteristic time is the inter-event interval τ . It turns out that τ itself depends on the level of subsampling: clearly, one is prone to wait longer for an event when measuring with a couple of electrodes than with hundreds of them. Therefore, using the authors' notation, if an event of size $X=s$ occurs in the fully sampled system, in the setup of neuronal avalanches it could well be “read” (or interpreted) as a *set* of smaller events of sizes s_1, s_2, \dots , because the binning of the data could introduce silent periods in between them. The *sum* of these s_k is being denoted as X_{sub} in the current manuscript. While an analytical understanding of $P(X_{\text{sub}})$ is certainly welcome, it would be highly desirable to attempt a connection with the more difficult, but also more relevant in some contexts, probability distribution of the smaller chunks $P(s_k)$. Note that the apparent exponent reported in Ref. [20] changes with the size of the time bin, a feature that is not captured in the framework of the present manuscript. I'd like the authors to comment on this problem, particularly in relation to previous references that have discussed it.

What is the status of a and b , as in “a family of scaling relations [defined by a and b] follows for power-laws” (line 470 of the SI)? Are they free parameters, can they be obtained from the data, or what else? This should be carefully explained in the main text.

Section 2.2 shows results for three models: a branching model sitting on an fully connected network, and a two-dimensional conservative sandpile model with either open or periodic boundary conditions. First, it is important to point out that the avalanche size exponent for the first is different from the last two, a fact that should be discussed (particularly in light of the statement on line 79, which assumes $\gamma > 1$). The exponent of the sandpile model is only mentioned en passant in line 160 in an attempt to explain the absence of deviations from the scaling prediction (“hairs”). Could the authors expand that explanation? Please state the exponent values of the models in the slopes of Fig. 2 (accompanied by the appropriate citations, since they are well known). In these models, the scaling proposed by the authors seem to work very nicely, which is reassuring. However, can they reproduce e.g. the results of the model in Ref. [10], which is very similar to a branching model, except that it occurs in a topology with finite average degree (random and small world graphs included)? Both a random graph and a fully connected graph give rise to the same mean-field avalanche exponent, but arguably fully connected graphs are biologically less realistic and typically have more pathological scaling issues (a crucial point since this is a manuscript about scaling relations). Note also that the literature consensually regards the power laws of the sandpile model to be strictly dependent on its microscopically conservative dynamics (open boundary conditions notwithstanding) which, from a biological perspective, is another highly non-realistic ingredient. So, to put it more directly, can the authors provide a microscopically non-conservative model on a not-fully-connected graph which yields the exponent observed in neuronal avalanches when subsampled? That would be a major result, without which a much deeper discussion of the limitations of the scaling results would be necessary. If the theory does not work in that more biologically acceptable scenario, why is that?

Please note that the above discussion affects section 2.3 as well, where the authors refer to “subcritical systems” in general, but in practice apply the theory to the branching model on a fully connected graph only. Once more, it would be necessary to probe the authors’ theory in a branching model with finite connectivity. In this context, I wonder whether the analytical insights in Larremore et al. PRE 85 066131 (2012) for the statistical properties of avalanches in networks could come to the authors’ help. Please comment.

Section 2.4 is very interesting in that the scaling analysis proposed by the authors is applied to real data. However, insofar as there is no clear consensus as to which model best reproduces the data (as far as I’m aware), the strength of the conclusions drawn from the data hinges on the robustness of the scaling analysis with respect to different models (which brings us back to the comments above). Please add the value of the observed exponents to Fig. 3. Moreover, both the text and abstract seem to suggest that mature cultures are generally critical, but a close inspection of Fig. S6 reveal supercritical bumps in $P(s)$ in about 50% of the experiments (day 28 and day 35). Stating this explicitly does not cloud, in my opinion, the beauty of the experimental collapses of those cultures which are presumably critical. More problematic, on the other hand, is the fact that the scaling seems to work well *even for the supercritical cases*. Could the authors elaborate on that?

I find that section 2.5 has some pedagogical value, in the sense that many physicists immediately think of finite-size effects when faced with the issue of subsampling for the first time. So making the distinction is certainly important. But finite-size scaling is, as the name suggests, a scaling theory, and a very well established one. Isn’t it possible to connect those

long-known scaling relations with the ones proposed for subsampling, and then test them for the models? As it stands, section 2.5 limits itself to showing numerically that changes in system size or sampling size have different effects. Can't this be extended to an analytical approach, as done previously in the manuscript?

Finally, a careful reading of the calculations in the Supplementary Material suggests that the analytical part of the current manuscript relies rather heavily on Ref. [3]. For the authors' own benefit, it would be important that the readers find somewhere in the paper a brief summary of the advances that were made here in comparison to that reference.

In summary, I believe this manuscript makes significant progress towards an analytical understanding of the problem of subsampling. However, the models used by the authors to test the theory need to be expanded, including at least branching models on random graphs with finite connectivity (even if only with annealed, finite-average-degree connectivity). These are still safely in the mean-field realm where the authors' method resides and is expected to work, as they explicitly stated in the discussion. In my opinion, results of this test would be interesting either way. If their scaling works there, do the results contradict those of Ref. [10]? If it does not, then why not?

Minor points:

- I believe the abstract is mostly appropriate, except perhaps for the sentence below,

“we derive analytically a subsampling scaling framework that is applicable to different observables, including distributions of ... node degrees, and of cluster sizes.”

where the authors mention “node degrees” and “cluster size”, which I'm not sure would be obvious to a broader audience. So it'd help if authors could put them in context. This issue reappears in lines 31-32 of the introduction. The reference to presumably critical experimental cultures also deserves some amendment, as mentioned previously.

- Fig. 1: What do the points represent? Simulations with actual subsampling or just a plot of the analytical solutions? Please clarify.

- Fig. 4: the horizontal log scale of the plots is difficult to understand

Regarding the Supplementary Material:

- I couldn't find the references for the Supplementary Information, so I'm understanding the numbers refer to the main bibliography (please confirm).

- In the deduction of Eq. S1 (and in the following equations as well) I'd suggest a change of notation to clarify the two arguments of G_{sub} (z and p), since it's constantly compared to the single-argument $G(z)$. Perhaps $G_{\text{sub}}(z;p)$?

- In SI 1.2, the normalization constant of the power-law distributions is written as $C_{\text{gamma}} = 1/\zeta(\text{gamma})$. Please define ζ (presumably the Riemann zeta function, as defined

only later in SI 2).

- Lines 485-487 could be improved to eliminate two consecutive “Therefore”s.

- Following the sequence of the main text, shouldn't Fig. S6 come before Fig. S5?

We want to thank both reviewers for raising important questions in their detailed comments. In our revisions, we have now addressed all of them and supplemented the manuscript accordingly. Our replies here are marked in blue.

The major changes are:

1. Improved definitions in the mathematical subsampling section.
2. Changed Figure 1, adding numerical subsampling results to it.
3. Expanding the discussion on the power-law exponents in the critical models.
4. Addition of two models, a sparsely connected branching network, and an integrate-and-fire model (EHE). Both are discussed in details in SI and referenced to in the text.
5. Addition of a detailed discussion of the binning in analysis of the neural recordings and models, including a new panel in Fig. 3 and a specially dedicated section in the SI.
6. Discussion of “bumps”, i.e. specific deviation from power law distributions observed in some of the experimental recordings.
7. Development of a mathematical formalism that combines subsampling scaling with finite-size scaling.
8. Development of a procedure for numerical optimization of subsampling scaling and verification that the analytical results coincide with the numerical ones.

There are also some minor changes improving the clarity and readability of the manuscript. All significant changes and additions are highlighted in the text (blue color). We include the exact line numbers in the reply to the referees.

In the following we address point by point the comments of the referees.

Reviewer #1 (Remarks to the Author):

In this manuscript, Levina and Priesemann aim to address the problem of subsampling, which has plagued the field of neuronal avalanches for many years. Priesemann has previously published on this, but the present work with Levina gives a firm analytical foundation to this problem and is therefore an important new contribution. Their results are convincing and demonstrate to my satisfaction that they have made great progress in this area, coming up with the correct equations to describe how power law and exponential distributions will be affected by subsampling. Based on this method, they go on to show that data collapse of the avalanche size distribution will occur only in models that are actually tuned to be critical, and that such collapse will not occur in models that are not critical. They also clearly demonstrate how the problem of subsampling is different from the problem of finite sized data. Finally, they apply their method to a publicly-available data set of cultured networks (from Potter’s group) and show that mature cultures show collapse, while young and still developing cultures do not.

I think that this is a very important new contribution, and clearly merits publication in Nature Communications for two main reasons. First, this problem of subsampling has been known for about a decade, but no clear solution has been presented until now. Here, this controversy is nearly solved, at least for some of the main cases (power law and exponential distributions). Second, this area of criticality in biological systems is receiving increased attention and is not confined merely to neuronal avalanches.

Criticality has been suggested in bacterial colonies, flocks of birds, and even human interactions. This work is thus very broadly applicable and is likely to be well-cited.

Thank you very much for the clear summary and for your comments on our manuscript. Below we reply to each comment and list in detail all changes and extensions we made to the manuscript.

That said, I have a few comments, questions and suggestions:

The problem of subsampling that you address is confined to limited spatial sampling (e.g., not having enough electrodes to record all the neurons in the population). This is an enormous problem and tackling it is a commendable task. But could you at least say a few words about how this problem is related to two other issues? I am thinking of recording length and temporal binning. Your ability to properly correct for subsampling assumes that the data were collected over a long period of time (at least one hour), and this may be limited in many preparations (those that rely on Ca^{++} imaging, for example in neuroscience). Must recordings be of some length (assuming some firing rate) for all of this to work?

Our work addressed the question how spatial subsampling affects the cluster size distributions. It assumes that a sufficient number of samples has been taken (e.g. sufficient recording length), that the observed samples have been drawn randomly (i.e. in an unbiased manner), and it assumes an unambiguous definition of the cluster, being it the node degree or the avalanche. These are important requirements and are now stressed more clearly in the main text (line 77, line 243).

The length of the recording constrains the quality of the reconstructed distribution in the observed (already subsampled) system. Indeed, if there are too few data points (avalanches), not much can be said about the distribution. But as soon as there is enough data to obtain a reasonable estimation for the distribution of the subsampled system, the whole theory will apply. Typically this requires at least a few thousand avalanches. In the experimental data we analyzed, there were at least 10,000 avalanches, typically a few 100,000 avalanches, thus more than sufficient to obtain smooth distributions over a three orders of magnitude.

Somewhat related to this, the temporal binning of the data you used was at 1 ms. But many labs bin at 5 ms, or even longer bins. Can you comment on how these will affect your conclusions on subsampling? In some sense, both of these issues are related to subsampling because they are also presenting limited views of what is actually going on in the data.

Indeed, in many experiments the bin size has been found to affect the avalanche size distribution (e.g. Beggs & Plenz, 2003, Priesemann et al. 2013]). This is because experimental systems often seem to lack a true separation of time scales leading to avalanches blending into each other or potentially being cut, and consequently the avalanche extraction via binning is ambiguous both in full system and under subsampling (see Priesemann et al. 2014 for an in depth discussion).

Interestingly, the in vitro spike data we evaluated here were fairly invariant against changes in bin size, when showing power laws, i.e. despite a 32 fold change in bin size, the slope remained unchanged (see Fig. 3D, which we added). This was the case for all experiments showing power laws, and indicated the presence of a separation of time scales (STS). A similar invariance was also reported for cortical slices [Friedman et al. PRL 2012, Fig. S1 in their supplementary material]. Hence in this experiment, binning had little impact on the collapse under subsampling scaling. This suggests that here the avalanche

definition is unambiguous in a large range of bin sizes. We added a figure panel to demonstrate this and also discuss the potential ambiguities of avalanche definitions (Fig. 3 D, lines 243, Suppl. Information SI 5).

When doing data collapse, there are generally two ways that one could proceed. First, you can get the analytically expected parameters (as you did) and then show that they produce a reasonable collapse when inspected by eye. Second, you can conduct a collapse by minimizing some type of error after searching through different parameter values. In this manuscript, you did the former but not the latter. I see that your plots look “pretty good” but I would also like to see some type of independent corroboration of their quality. Can you show that deviating from the parameter values you have chosen will actually increase the distance between the curves? There are now some toolboxes out there that claim to do automated collapse by minimization of error.

Thank you for this suggestion. It is indeed a logical, independent corroboration. We implemented a numerical procedure similar to that of Marchall et al. 2016, with an additional step of transforming the distribution first to a log-log space. This step is necessary to not underweight the tails. We used the L1 norm to avoid an over-weighting of large deviations generated by the small number of samples in the (more noisy) tail. Our numerical results are in agreement with the analytic solution. We added a detailed section to the SI 8 (“Numerical estimation of optimal scaling”) and refer to the results in the main text (lines 298).

Minor:

In supplementary figure 1, what is ASMC? Don't you mean BTWC? Or BTW with periodic boundary Conditions?

Thank you for pointing this out. We indeed meant to write BTWC, not ASM (“Abelian Sandpile Model”), which is an alternative name of the BTW we used earlier.

In the supplementary figures, it looks like there is a “supercritical bump” in some of the files, even though they show what appears to be good collapse. Can you explain this apparent discrepancy?

This is indeed an important observation, which we omitted to discuss in the previous version of the manuscript, and we thank the reviewer for pointing this out. We added a whole section to the SI and summarize it in the results (line 255, and SI 6 “Detailed discussion of the experimental results”). We kindly refer the reviewer to that section. In brief, at first glance, the bumps are reminiscent of supercritical systems. However, supercritical neural models typically show bumps at system or sampling size ($s = N$), not at those very large sizes as here ($s = 5.000$). We thus suggest that the bumps are more likely to originate from neurophysiological finite size effects (see SI 6 for the full line of argumentation).

Reviewer #2 (Remarks to the Author):

I have read with interest the manuscript by A. Levina and V. Priesemann. It brings an interesting analytical understanding to the problem of subsampling, which has deservedly received considerable attention in the literature of neuronal avalanches. Specifically, the paper assumes that events X originally described by a probability distribution $P(X)$ are sampled with a probability p , rendering events X_{sub} whose effective probability distribution is $P_{\text{sub}}(X_{\text{sub}})$. Scaling relations relating P_{sub} to P and p are derived, some of which had been previously published by Stumpf et al PNAS 2005 (Ref. [3] of the manuscript). I find that the paper is relevant and mostly well written. There are, however, essential points that I would like to see discussed before the manuscript could be considered suitable for publication in Nature Communications.

Thank you very much for the clear summary and for thoroughly reviewing of our manuscript. Addressing the issues you raised significantly improved our manuscript. Below we list in detail all changes and reply to the comments.

To start with, I suggest the authors provide simple examples of the mathematical quantities upon which they are going to develop all their calculations (N_0 , N , X , X_{sub} etc). One can understand them in retrospect, once the calculations are advanced, but the manuscript could read much better and gain much more traction if the description of the problem was clearer from the start. Lines 59 and 64 have unclear expressions, such as " N_0 that represents the number of events in a cluster", or "each event or component of every cluster is independently observed". Since a "cluster" has not been properly defined, the above sentences, as they stand, hinder more than they help. Perhaps one simple example preceding them would suffice.

Thank you for pointing this out. We added more detailed explanation, including examples, to the main text as proposed (line 65, and line 75). More technical expressions were moved to the supplementary information.

Most importantly, the authors' mathematical description of subsampling leave aside, as far as I can understand, a very important aspect of subsampling, namely, the time-dependent nature of the underlying processes. Consider, for instance, neuronal avalanches, whose sizes are defined as the number of events (e.g. local field potential large-enough deviations) that occur in between silent epochs. To properly define what a silent epoch is, data is usually binned using some characteristic time of the system. In Ref. [20], for instance, this characteristic time is the inter-event interval τ . It turns out that τ itself depends on the level of subsampling: clearly, one is prone to wait longer for an event when measuring with a couple of electrodes than with hundreds of them. Therefore, using the authors' notation, if an event of size $X=s$ occurs in the fully sampled system, in the setup of neuronal avalanches it could well be "read" (or interpreted) as a *set* of smaller events of sizes s_1, s_2, \dots , because the binning of the data could introduce silent periods in between them. The *sum of these s_k * is being denoted as X_{sub} in the current manuscript. While an analytical understanding of $P(X_{\text{sub}})$ is certainly welcome, it would be highly desirable to attempt a connection with the more difficult, but also more relevant in some contexts, probability distribution of the smaller chunks $P(s_k)$.

We agree that the issue of an unambiguous avalanche extraction from neural recordings, e.g. by binning, is crucial and subsampling further complicates it. And indeed, for our derivation of scaling under subsampling, we assume that an avalanche of the full system translates to exactly one avalanche (potentially of size zero) under subsampling. We added a statement about this to the subsampling definition, so that this point is not missed (line 77, and in more detail in line 243, and SI 5 “Avalanche definition, binning, and subsampling scaling”).

The main aim of this paper, however, is to derive how to infer $P(s)$ of the full system under subsampling. Derive how ‘cutting’ of one avalanche and potentially also ‘merging’ of subsequent avalanches affects the distribution it is a whole new topic of large importance. We believe that it requires an independent paper, because a theory of cutting and merging involves a number of parameters and assumptions: The cutting behavior depends on the avalanche shape, the merging on the inter-avalanche-intervals, and all these results will depend on the choice of bin size. We will certainly tackle this project as a next step, but for now would like to concentrate on the main result of the current paper, namely how to infer $P(s)$ of the full system under subsampling - assuming an unambiguous avalanche or cluster definition. If the more appropriate avalanche extraction mechanism will be found, our results can be applied without any changes.

In models, avalanches can easily be defined unambiguously by making use of the separation of time scales (STS), which is often seen as necessary condition for self-organized criticality (e.g. Dickman et al. Braz. J. of Phys. 2000): Pauses introduced by subsampling are typically much shorter than those between avalanches. As a consequence, intermediate bin sizes do not cut or merge avalanches. This invariance of avalanche distributions against changes in bin size thus can indicate a STS.

Interestingly, the in vitro spike data we evaluated here were fairly invariant against changes in bin size, when showing power laws, i.e. despite a 32 fold change in bin size, the slope remained unchanged (see figure 3D, which we added). This was the case for all experiments showing power laws, and indicated the presence of a separation of time scales (STS). A similar invariance was also reported for cortical slices [Friedman et al. PRL 2012, supplementary material Fig. S1]. Hence in this experiment, binning had little impact on the collapse under subsampling scaling. This suggests that here the avalanche definition is unambiguous in a large range of bin sizes. We added a figure panel to demonstrate this and discuss the potential ambiguities with avalanche definitions (Fig. 3D, line 243, and SI 5 “Avalanche definition, binning, and subsampling scaling”).

Note that the apparent exponent reported in Ref. [20] changes with the size of the time bin, a feature that is not captured in the framework of the present manuscript. I’d like the authors to comment on this problem, particularly in relation to previous references that have discussed it.

We investigated the impact of the bin size on the power-law slope and found that in vitro spike data we evaluated were fairly invariant against changes in bin size, probably because of an underlying STS (see above): when showing power laws, the slope remained unchanged, despite a 32 fold change in bin size (see figure 3D, which we added).

For the models, which have a STS, we observed exactly the same effect: With very small bin sizes the distributions still changed, but with sufficiently large bin sizes the distributions were invariant and showed exactly the same power law slope as the fully sampled system. We added a figure to demonstrate this (see Figure S7A, SI 5 “Avalanche definition, binning, and subsampling scaling”).

What is the status of a and b , as in “a family of scaling relations [defined by a and b] follows for power-laws” (line 470 of the SI)? Are they free parameters, can they be obtained from the data, or what else? This should be carefully explained in the main text.

Indeed, if $P(s)$ follows a perfect power law (i.e. without cutoff) one of the variables, a or b , can be chosen freely. The other variable is constrained by the relation between a , b and γ (Eq. 4). We made this more clear in the main text and the SI (line 125, and SI around equation SI 14).

Section 2.2 shows results for three models: a branching model sitting on a fully connected network, and a two-dimensional conservative sandpile model with either open or periodic boundary conditions. First, it is important to point out that the avalanche size exponent for the first is different from the last two, a fact that should be discussed (particularly in light of the statement on line 79, which assumes $\gamma > 1$). The exponent of the sandpile model is only mentioned en passant in line 160 in an attempt to explain the absence of deviations from the scaling prediction (“hairs”). Could the authors expand that explanation? Please state the exponent values of the models in the slopes of Fig. 2 (accompanied by the appropriate citations, since they are well known).

Indeed, it is precisely because of the different exponents of the 2D BTW and BM that we chose these different models. We now state the exponents explicitly, including references, in the main text (lines 157, specifically line 169 and 181). We also derive the absence of the “hairs” for power laws with slope approaching unity from above adding exact computations to the SI (SI 1.3 “Power-law exponent close to unity”).

In these models, the scaling proposed by the authors seems to work very nicely, which is reassuring. However, can they reproduce e.g. the results of the model in Ref. [10], which is very similar to a branching model, except that it occurs in a topology with finite average degree (random and small world graphs included)?

This is an important and interesting question, thank you for raising it. Indeed, Ribeiro et al. study a network that is very similar to one of our examples, the branching network, and vary the topology from local via a specific small-world like topology to random, all with separation of time scales. The crucial difference between our models is the binning for defining the avalanches. As you pointed out earlier, selecting very small bin size (in the study of Ribeiro et al it was fixed to one time step, the smallest time constant in the network) can cut avalanches under subsampling, resulting in the apparent subcritical behavior. We now reproduced their approach, i.e. using a bin size of one step, using the BM with sparse, annealed topology, and found the same results (Fig. S7C, section SI 5 “Avalanche definition, binning, and subsampling scaling”).

Importantly, in the models only a sufficiently large bin-size recovers the critical distribution (because of avoiding cutting avalanches, see above), and allows for a collapse under subsampling scaling (Fig. S7).

For this study, we were particularly interested in the time-scale separated case to disentangle the avalanche extraction problem from the subsampling problem and thus have a ground truth solution for perfectly extracted avalanches, and more general clusters (e.g. node degrees). We added a discussion

about the difference to the Ribeiro et al. (SI 5 “Avalanche definition, binning, and subsampling scaling”). We also demonstrated that our results hold for the sparse randomly connected branching model (see corresponding replies below).

Both a random graph and a fully connected graph give rise to the same mean-field avalanche exponent, but arguably fully connected graphs are biologically less realistic and typically have more pathological scaling issues (a crucial point since this is a manuscript about scaling relations). Note also that the literature consensually regards the power laws of the sandpile model to be strictly dependent on its microscopically conservative dynamics (open boundary conditions notwithstanding) which, from a biological perspective, is another highly non-realistic ingredient. So, to put it more directly, can the authors provide a microscopically non-conservative model on a not-fully-connected graph which yields the exponent observed in neuronal avalanches when subsampled? That would be a major result, without which a much deeper discussion of the limitations of the scaling results would be necessary. If the theory does not work in that more biologically acceptable scenario, why is that?

Indeed, the random and fully connected networks are very similar in terms of the mean-field behavior and precisely this reasoning led us to use fully connected networks in the previous version of the manuscript. To demonstrate that deviations from a mean-field are not essential for our results, we extend the branching model, including also sparse random connectivity ($k=4$, annealed, Fig. S6). We additionally considered the EHE model (originally a non-leaky integrate-and-fire fully connected network, Eurich et al, PRE 2002), here implemented both in a fully-connected version as well as with 10% random network topology (section SI 4, Fig. S5). In all models, the subsampling scaling works as derived analytically, demonstrating its generality.

Whether the generation of power-law distributions is possible in “not local energy conserving” model (for example, for SOC) is a matter of a long-standing debate, and to our knowledge, there is no decisive point so far. Our study aims at understanding the subsampling effects in data and verifying it in the existing models. Developing a new model will be very ambitious goal outside of the scope of the present manuscript. However, we implemented a non-conservative model of integrate-and-fire neurons (Eurich et al, PRE 2002), which is truly critical in the conservative limit, but for any finite system size it dissipates energy with every spike. This model is known to produce power-law distributions for avalanche sizes with slope approximately -1.5 . We additionally modified the original, fully connected network to be 10% randomly connected. In both cases, subsampling scaling leads to a good collapse of the distributions. Our results are described in section 4 of the SI and mentioned in the main text (line 171).

Please note that the above discussion affects section 2.3 as well, where the authors refer to “subcritical systems” in general, but in practice apply the theory to the branching model on a fully connected graph only. Once more, it would be necessary to probe the authors’ theory in a branching model with finite connectivity. In this context, I wonder whether the analytical insights in Larremore et al. PRE 85 066131 (2012) for the statistical properties of avalanches in networks could come to the authors’ help. Please comment.

We performed additional simulations on a sparsely randomly connected branching model. We consider Markov matrix (sum of all outgoing connections is equal 1) and thus the largest eigenvalue of our matrix

is 1 leading to a critical network, in accord with Larremore et al. computations. As expected from the mean-field approximation, our approach worked the same on the random network as on the fully connected one. We include the results from the sparsely connected network in the SI (Fig. S6, section SI 4) and also discuss them in the main text (line 171).

Section 2.4 is very interesting in that the scaling analysis proposed by the authors is applied to real data. However, insofar as there is no clear consensus as to which model best reproduces the data (as far as I'm aware), the strength of the conclusions drawn from the data hinges on the robustness of the scaling analysis with respect to different models (which brings us back to the comments above).

There is indeed no "best model" for neuronal avalanches yet, despite more than 10 years of ongoing research. For this reason we investigated how our subsampling scaling performs on different models of different universality classes, to demonstrate the model independence and generality. In this resubmission we expanded the variety of the models, and now the manuscript shows results for the branching model on different topologies, the BTW with different boundary conditions, and a non-conservative integrate-and-fire network (EHE). Altogether, we showed that scaling approach is a model-independent view of the neuronal (or any other) subsampled data.

Please add the value of the observed exponents to Fig. 3. Moreover, both the text and abstract seem to suggest that mature cultures are generally critical, but a close inspection of Fig. S6 reveal supercritical bumps in $P(s)$ in about 50% of the experiments (day 28 and day 35). Stating this explicitly does not cloud, in my opinion, the beauty of the experimental collapses of those cultures which are presumably critical. More problematic, on the other hand, is the fact that the scaling seems to work well *even for the supercritical cases*. Could the authors elaborate on that?

We added the values of the exponents to Fig. 3, it is approximately -2 for all cultures exhibiting power law behavior. Concerning the bumps, this is an important observation, we thank the reviewer for pointing this out, and we expanded on the description of the results and SI (line 255, and SI 6 "Detailed discussion of the experimental results").

In the following we address the two questions, whether to expect a collapse here, and whether the bump indicates supercriticality. We added an in depth discussion, mainly identical with the detailed reply below, to the section SI 6, and a short summary of it the main text (line 255).

Regarding the questions whether the distributions observed here are expected to collapse, the answer is straight forward: The avalanches in the tail make only a tiny fraction of all observed avalanches (about 2 in 10.000), while the other 99.98 % avalanches follow a power law for about 3 orders of magnitude. (It is the log-log scale together with the logarithmic binning that might make the bumps appear more prominent than they are.) With only 0.02% of avalanches not following a power law, a decent collapse is to be expected for the power law. The collapse of the bumps itself is a manifestation of the activity spread during the large avalanches that hit the sampled set proportionally to the number of sampled units.

For the origin of the bumps, there are a number of potential explanations, which we discuss one after the other. The hypothesis that they indicate a slightly supercritical state is one of them.

We see different potential explanations for the bump appearance:

1) *Biological finite size effects in a critical system*

Assume the neural cultures were precisely at a critical point and the distribution of the avalanche sizes is a perfect power law without cutoff. Then the probability P_{bump} to observe an avalanche larger than s_{trans} is given by the Hurwitz zeta function. What happens in a biological system with these avalanches? Their size cannot go to infinity, because some biological mechanisms (like, for example depletion of synaptic resources, shortage of Ca^{2+} or even homeostatic mechanisms) would limit their maximal size. Thus all avalanches larger than some s_{trans} (in our data $s_{\text{trans}} = 3000$) are distributed around a characteristic, biologically determined size, which in case of our data is about $s = 5000$. In agreement with this hypothesis, the number of the avalanches observed in the “bump” agrees with the probability P_{bump} for perfectly critical system. Thus the data support our hypothesis that the bump represents the collection of all avalanches that would, in an ideal system, be larger than 3000. (Note that all sizes s given here are the sizes observed under subsampling).

2) *Criticality alternates with a state that gives rise to large avalanches.*

The in vitro neural networks we analyzed could in principle alternate between different states. While in one state, which comprises about 99.98 % of the avalanches, the system is critical, in the other state it displays unusually large avalanches that run multiple times over the entire system and give rise to population bursts, i.e. they manifest as the observed bumps. (Note that the bumps make only about 0.02 % of the avalanches and are of size $s \approx 5000$). How many “burst avalanches” are generated, might depend on the individual culture (some showing none at all), and it could be pure coincidence that the fraction of burst avalanches is in agreement with the fraction expected for the avalanche tail (see hypothesis 1).

3) *A novel form of slight supercriticality in a finite system.*

While it is straight forward to identify “subcriticality” (no avalanches covering the full system size, no power law behavior of distributions, but a prominent exponential tail), it is much trickier to identify “supercriticality” in neural systems by pure observation, potentially because supercriticality in the thermodynamic limit implies a non-zero fraction of infinite avalanches, but in finite systems it depends on the type of system how these infinite avalanches manifest. For supercritical systems in neuroscience, the bump in $P(s)$ occurs typically at N , i.e. the system size or the number of sampled neurons [e.g. Eurich 2002]. However, here in all experiments where the bump is observed, it is around 80 times N (i.e. $s \approx 5000$ from sampling up to 60 electrodes). In fact, a unit spikes about 80 times in a single of these large avalanches. Thus the bumps do not indicate supercritical behavior resembling that of previous studies. However, it could indicate a novel form of supercritical behavior on a finite system.

Distinguishing between these potential causes of the bump would need additional experimental investigations that are outside of the scope of present manuscript. The main goal of our work here is to provide a method to extrapolate distributions from a subsampled system to understand the behavior of the full system. An absence of a collapse under subsampling scaling makes it (currently) impossible to infer the distribution in the full system. In the experiments evaluated here, the presence of the data

collapse in the more mature networks predicts a power-law distribution for $P(s)$ of the full neural system that spans approximately 6 orders of magnitude. Whether such power laws are, however, sufficient to infer criticality, is under debate. Additional established methods, e.g. shape collapse (Friedman et al. PRL 2012), could help to further test this question.

I find that section 2.5 has some pedagogical value, in the sense that many physicists immediately think of finite-size effects when faced with the issue of subsampling for the first time. So making the distinction is certainly important. But finite-size scaling is, as the name suggests, a scaling theory, and a very well established one. Isn't it possible to connect those long-known scaling relations with the ones proposed for subsampling, and then test them for the models? As it stands, section 2.5 limits itself to showing numerically that changes in system size or sampling size have different effects. Can't this be extended to an analytical approach, as done previously in the manuscript?

Thank you for this comment. Indeed, it was important for us to emphasize the difference of subsampling scaling and finite-size scaling (FSS). Following your suggestion we extended our analysis and derived a combined subsampling-finite-size scaling that allows for critical systems to extrapolate from arbitrary subsampling of arbitrary system size to any other subsampling of any other system size. We also performed numerical simulations to test the obtained scaling relationship. We added a paragraph and a figure (Fig. 5) to the main text (section 2.5 "Subsampling versus finite size scaling", line 286) and an entire section treating the derivations to the SI (section SI 7 "Combining subsampling scaling and finite-size scaling").

Finally, a careful reading of the calculations in the Supplementary Material suggests that the analytical part of the current manuscript relies rather heavily on Ref. [3]. For the authors' own benefit, it would be important that the readers find somewhere in the paper a brief summary of the advances that were made here in comparison to that reference.

We agree that it is very important to attribute the results properly, and we are sorry that it has not been as clear as we intended. We expanded this topic now both in the main text and to the SI (line 52, 116, first paragraph in the SI). The main point of difference is that in [3] the question of which distribution change and which keep their classes under subsampling was investigated. We extend the study of Stumpf et al. [3] and develop formalism allowing to infer the original distribution from the subsampling, also in the case where an exact solution is not possible.

In summary, I believe this manuscript makes significant progress towards an analytical understanding of the problem of subsampling. However, the models used by the authors to test the theory need to be expanded, including at least branching models on random graphs with finite connectivity (even if only with annealed, finite-average-degree connectivity). These are still safely in the mean-field realm where the authors' method resides and is expected to work, as they explicitly stated in the discussion. In my opinion, results of this test would be interesting either way.

Thank you for this summary. We followed your suggestions and included additional models (the sparsely connected BM, and the 10% and fully connected EHE for its non-conservative dynamics). For all of them, as predicted by the analytical computations, scaling results stay unchanged.

If their scaling works there, do the results contradict those of Ref. [10]? If it does not, then why not?

We refer to our replies above. In brief: We get the same results as Ribeiro et al. when applying very small bin sizes (so that avalanches are cut). As a consequence, the subsampled system can be mistaken as subcritical, and thus scaling does not lead to a collapse. Our approach requires an unambiguous definition of avalanches (one avalanche in the fully system translates to one under subsampling).

Minor points:

- I believe the abstract is mostly appropriate, except perhaps for the sentence below,

“we derive analytically a subsampling scaling framework that is applicable to different observables, including distributions of ... node degrees, and of cluster sizes.”

where the authors mention “node degrees” and “cluster size”, which I’m not sure would be obvious to a broader audience. So it’d help if authors could put them in context. This issue reappears in lines 31-32 of the introduction. The reference to presumably critical experimental cultures also deserves some amendment, as mentioned previously.

Thank you very much for the comment. We now removed the term “cluster” in abstract and introduction, because it is a term used in different contexts without a clear definition. We then introduce it in the results section thoroughly. Regarding the term “degree” of the node, it is mentioned as an example of additional application area and its understanding is not necessary for the understanding of the article, however we believe it is sufficiently well known and it broadens set of potential readers.

- Fig. 1: What do the points represent? Simulations with actual subsampling or just a plot of the analytical solutions? Please clarify.

Indeed, in the original submission we plotted exact analytical solutions. Following your question, we decided to add direct numerical simulations. Now points represent the numerical simulations and lines analytical solutions, a small jitter originated from finite numerical sampling is visible in the numerical part. We changed the Figure 1 and it’s caption to make the data origin clear.

- Fig. 4: the horizontal log scale of the plots is difficult to understand

We have reduced the number of ticks in all figures of the main manuscript and hope that now the axis scale is easier to read.

Regarding the Supplementary Material:

- I couldn't find the references for the Supplementary Information, so I'm understanding the numbers refer to the main bibliography (please confirm).

This is correct. We now prepared a separate bibliography for the SI.

- In the deduction of Eq. S1 (and in the following equations as well) I'd suggest a change of notation to clarify the two arguments of $G_{\text{sub}}(z \text{ and } p)$, since it's constantly compared to the single-argument $G(z)$. Perhaps $G_{\text{sub}}(z;p)$?

Thank you, we changed the notation as you suggested.

- In SI 1.2, the normalization constant of the power-law distributions is written as $C_{\text{gamma}} = 1/\zeta(\text{gamma})$. Please define ζ (presumably the Riemann zeta function, as defined only later in SI 2).

Yes, it is indeed the Riemann zeta function. We now specify it early on in the text (SI 1.2).

- Lines 485-487 could be improved to eliminate two consecutive "Therefore"s.

Thank you for noticing, we improved the style and restructured this part of the text.

- Following the sequence of the main text, shouldn't Fig. S6 come before Fig. S5?

We now ordered all figures and all the SI according to their appearance in the main text. The only exception is SI 5, which is mentioned briefly in the beginning of the results, but is of importance only later.

Reviewers' comments:

Reviewer #1 (Remarks to the Author):

I find that the authors have done a good job of responding to the issues that I have raised. I appreciate their efforts to verify the analytical exponents numerically, and the results were encouraging. The more extensive comments by reviewer 2 also seemed to be addressed. The process of accounting for combined finite size and subsampling effects is very nice and adds value to the manuscript. Overall, this is a timely and accurate piece of work.

Reviewer #2 (Remarks to the Author):

The revised version of the manuscript shows considerable improvement over the previous one. Most of the questions posed have been answered. There are, however, some issues that remain and others that have emerged in light of their response. Those should be addressed before I could consider the manuscript acceptable for publication.

First, I believe the manuscript would read better if authors included slopes with the values of the exponents in all the figures where power laws appear. That includes figures 2, 5, S1, S5, S6, S7 and S8. In Fig. 4, the exponent in the vertical axis is unclear (it looks like a ν , but it might well be a γ) and its value should be added to the figure caption (I could only find it in the SI).

Second, I've noticed that notation can be confusing at times. Consider, for instance, Fig. 2A. If s is the size of avalanches in the subsampled system, shouldn't the vertical axis be P_{sub} instead of $p(s)$? (incidentally, I see $p(s)$ instead of $P(s)$ with a capital "P", isn't that confusing with sampling parameter p ?). I suspect this confusion in notation ($P_{\text{sub}}(s)$ versus $P(s)$) recurs throughout the manuscript, so I'd urge authors to check carefully and amend figures accordingly.

Third, and most importantly, I find that the discussion about the bin size in Section SI 5 is too interesting and important to be relegated to the supplementary material. I would therefore encourage the authors to move at least part of it to the main text. Figure S7, for instance, is particularly interesting in that it helps understand Fig. 3. In regard to that discussion, I also have a few questions:

- The issue of bin size for the branching model is focussed on the sparse connectivity variant. But does it occur as well in the fully connected variant? Please comment. For instance, I didn't find the bin size used to obtain the avalanches for the (fully connected) branching model of Figs. 2, 4 and 5. Please clarify. Bin size information is also missing from Fig. S6. Since it precedes the discussion presented in Fig. S7, it gets confusing (after reading SI 5 and appreciating Fig. S7, I went back to Fig. S6 but didn't find the bin size).
- The paper also lacks an explicit comparison between the results of Fig. S7 and those of

Ref. [9], whose claims it directly addresses. The authors should not miss the opportunity to advance a discussion raised in the literature.

- Finally, in the context of Fig. 3D the authors mention that “the approximate invariance of $P(s)$ against changes in the bin size indicates a separation of time scales in the experimental preparation”. In apparent contradiction with that claim, however, Ref. [12] displays in its Fig. 3 a (now famous) family of $P(s)$ curves with the power law exponent changing continuously with the value of the bin size. Could the authors comment on that difference?

I commend the authors for their notable effort in improving the manuscript. I believe that they will be able to satisfactorily address the points above.

As one last remark, there are several minor corrections that I list below for the authors' considerations:

Line 88: Authors still refer to N_0 , which has not been defined in the main text. N_0 reappears still undefined in line 104. I know what N_0 is, but in the spirit of a journal with a general audience, it wouldn't hurt to define it.

Line 102: “... is not a power law but only approaching it in the limit of...”

Line 107, Eq. (2): I assume the logarithm is in the natural base, but that could be made explicit (or perhaps use “ln” instead)

Caption of Fig. 3: “In panels A, B, and C bin size 1 ms”

Fig. 3: for consistency with Fig. S8, please add “c” to the vertical axis (and define it).

Lines 269 and 270: remove comma from “... data affected by both, subsampling and finite system size effects”

Supplementary information:

Line 1: “... we in detail derive ...”

Lines 94 and 95: “... and make use the geometric series”

Line 175: remove comma from “... both, the classical fully...”

Line 216: “... to demonstrated the impact ...”

Line 307: “... subsampling sclaing relations”

Line 336: remove comma from “... both, the BM and...”

We thank both reviewers for their kind remarks and questions. We are convinced that our combined efforts greatly improved the manuscript making it now ready for publication.

In the present revision we addressed all the comments of the Reviewer 2: moved the section 5 from supplementary to the main text, added one panel to the figure and did minor corrections. All significant changes are highlighted in the text (blue color).

Reviewer #1 (Remarks to the Author):

I find that the authors have done a good job of responding to the issues that I have raised. I appreciate their efforts to verify the analytical exponents numerically, and the results were encouraging. The more extensive comments by reviewer 2 also seemed to be addressed. The process of accounting for combined finite size and subsampling effects is very nice and adds value to the manuscript. Overall, this is a timely and accurate piece of work.

We thank the referee for the kind comments and support in improving the manuscript.

Reviewer #2 (Remarks to the Author):

The revised version of the manuscript shows considerable improvement over the previous one. Most of the questions posed have been answered. There are, however, some issues that remain and others that have emerged in light of their response. Those should be addressed before I could consider the manuscript acceptable for publication.

Thank you for the encouragement. We revised the manuscript in accordance with your comments and questions.

First, I believe the manuscript would read better if authors included slopes with the values of the exponents in all the figures where power laws appear. That includes figures 2, 5, S1, S5, S6, S7 and S8. We included slopes for all the figures mentioned, either as a dashed line in the figure, or if there was no sufficient space available, we denoted the slope in the figure legend.

In Fig. 4, the exponent in the vertical axis is unclear (it looks like a ν , but it might well be a γ) and its value should be added to the figure caption (I could only find it in the SI).

Thank you. We changed the font and now it is clear that we mean γ (i.e. the power-law exponent). We also added to the figure caption the values of γ for both models. (Note, that now it is Fig. 5)

Second, I've noticed that notation can be confusing at times. Consider, for instance, Fig. 2A. If s is the size of avalanches in the subsampled system, shouldn't the vertical axis be P_{sub} instead of $p(s)$? (incidentally, I see $p(s)$ instead of $P(s)$ with a capital "P", isn't that confusing with sampling parameter

p?). I suspect this confusion in notation ($P_{\text{sub}}(s)$ versus $P(s)$) recurs throughout the manuscript, so I'd urge authors to check carefully and amend figures accordingly.

Thank you! Indeed, the y-label in Fig. 2 should have been $P(s)$. Before, we had used $P(s)$ instead of $P_{\text{sub}}(s)$ whenever the distinction was not of importance. Now, to improve the consistency, we changed the $P(s)$ to $P_{\text{sub}}(s)$ in all figures, figure legends, and in the text, wherever appropriate.

Third, and most importantly, I find that the discussion about the bin size in Section SI 5 is too interesting and important to be relegated to the supplementary material. I would therefore encourage the authors to move at least part of it to the main text. Figure S7, for instance, is particularly interesting in that it helps understand Fig. 3. In regard to that discussion, I also have a few questions:

We transferred the former section SI 5 and Figure S7 to the main text (now section 2.4 and figure 3, respectively). The text of SI 5 was kept almost unchanged, apart from reordering the paragraphs to better fit the section into the flow of the main text. All significant changes are marked blue.

The issue of bin size for the branching model is focused on the sparse connectivity variant. But does it occur as well in the fully connected variant? Please comment.

Yes, the same principle holds for the fully connected BM. We added a figure panel (Fig. 3 D) to demonstrate this.

For instance, I didn't find the bin size used to obtain the avalanches for the (fully connected) branching model of Figs. 2, 4 and 5. Please clarify. Bin size information is also missing from Fig. S6. Since it precedes the discussion presented in Fig. S7, it gets confusing (after reading SI 5 and appreciating Fig. S7, I went back to Fig. S6 but didn't find the bin size).

All the simulated models were implemented with a separation of time scales (see section 2.4, the smallest inter-avalanche-interval (IAI) is longer than the longest avalanche). Hence any bin size that is larger than the longest avalanche can be used for an unambiguous definition of clusters or avalanches under arbitrary subsampling. Thus in all the figures you mentioned, the results hold for any bin size with $\max(\text{avalanche duration}) < \text{bin size} < \min(\text{IAI})$.

We added this information about the choice of bin size now to the methods ("4.2.4 Avalanche extraction in the models") and expanded on the definition in the main text (line 79 "For neural avalanches...").

The paper also lacks an explicit comparison between the results of Fig. S7 and those of Ref. [9], whose claims it directly addresses. The authors should not miss the opportunity to advance a discussion raised in the literature.

We now explicitly refer to the results by Ribeiro et al [9], mentioning that because of a too small bin size (bin size 1 time step), the critical BM under subsampling did not show a power law (line 268). The same holds for our own previous study (Priesemann et al. 2014).

Finally, in the context of Fig. 3D the authors mention that "the approximate invariance of $P(s)$ against changes in the bin size indicates a separation of time scales in the experimental preparation". In apparent contradiction with that claim, however, Ref. [12] displays in its Fig. 3 a (now famous) family of $P(s)$ curves with the power law exponent changing continuously with the value of the bin size. Could the authors comment on that difference?

Indeed, a change in $P(s)$ with bin size is often observed for neural avalanche distributions obtained from local field potentials (LFP) both in Ref.[12] Beggs & Plenz 2003, and in many subsequent studies. The data we analyze here is from spikes in developing cultures. Our results show that this neural system has an approximate separation of time scales (STS). One probable explanation would be that LFP recordings have less STS. The study of the possible causes of less STS in LFP recordings would be an interesting topic for a separate research project that we are planning to pursue. We now discuss this issue briefly in the main text (line 257).

I commend the authors for their notable effort in improving the manuscript. I believe that they will be able to satisfactorily address the points above.

We thank the reviewer for the time and efforts spent on helping us to improve the manuscript

As one last remark, there are several minor corrections that I list below for the authors' considerations:

Line 88: Authors still refer to N_0 , which has not been defined in the main text. N_0 reappears still undefined in line 104. I know what N_0 is, but in the spirit of a journal with a general audience, it wouldn't hurt to define it.

We totally agree, we added the definition of N_0 at the first appearance point.

Line 102: "... is not a power law but only approaching it in the limit of..."

Thank you.

Line 107, Eq. (2): I assume the logarithm is in the natural base, but that could be made explicit (or perhaps use "ln" instead)

To avoid confusions, we replaced all "log" by "ln"

Caption of Fig. 3: "In panels A, B, and C bin size 1 ms"

Thank you. We added the sentence to the figure legend.

Fig. 3: for consistency with Fig. S8, please add "c" to the vertical axis (and define it).

Thank you. "c" denotes the total number of avalanches when sampling with all electrodes. We have added this information, and the precise number of avalanches, to the figure legend.

Lines 269 and 270: remove comma from "... data affected by both, subsampling and finite system size effects"

Thank you.